# Epigenetic Targeting of Histone Deacetylases in Diagnostics and Treatment of Depression

**DOI:** 10.3390/ijms22105398

**Published:** 2021-05-20

**Authors:** Hyun-Sun Park, Jongmin Kim, Seong Hoon Ahn, Hong-Yeoul Ryu

**Affiliations:** 1Department of Biochemistry, Inje University College of Medicine, Busan 47392, Korea; 2Division of Biological Sciences, Sookmyung Women’s University, Seoul 04310, Korea; jkim@sookmyung.ac.kr; 3Research Institute for Women’s Health, Sookmyung Women’s University, Seoul 04310, Korea; 4Department of Molecular and Life Science, College of Science and Convergence Technology, Hanyang University ERICA Campus, Ansan 15588, Korea; hoon320@hanyang.ac.kr; 5BK21 FOUR KNU Creative BioResearch Group, School of Life Sciences, College of National Sciences, Kyungpook National University, Daegu 41566, Korea

**Keywords:** histone deacetylase (HDAC), depression, biomarker, anti-depressant therapy

## Abstract

Depression is a highly prevalent, disabling, and often chronic illness that places substantial burdens on patients, families, healthcare systems, and the economy. A substantial minority of patients are unresponsive to current therapies, so there is an urgent need to develop more broadly effective, accessible, and tolerable therapies. Pharmacological regulation of histone acetylation level has been investigated as one potential clinical strategy. Histone acetylation status is considered a potential diagnostic biomarker for depression, while inhibitors of histone deacetylases (HDACs) have garnered interest as novel therapeutics. This review describes recent advances in our knowledge of histone acetylation status in depression and the therapeutic potential of HDAC inhibitors.

## 1. Introduction

Depression is characterized by recurrent episodes of sadness and despondency (depressed mood) frequently accompanied by anhedonia, loss of appetite, reduced concentration and energy, excessive guilt, and recurrent suicidal ideation [1]. Despite treatment, more than 50% of patients experience recurrent episodes and approximately 80% of those with a history of two episodes experience another relapse [2]. Both the incidence and prevalence of depression are increasing, and depression is now a major global healthcare burden and cause of lost economic productivity [3]. Current treatment guidelines recommend modulators of monoaminergic transmission such as monoamine oxidase (MAO) inhibitors and specific serotonin reuptake inhibitors (SSRIs) as first-line therapy based on the theory that depression arises from abnormal monoaminergic transmission. However, despite the availability of many monoamine modulators, approximately 50% of patients are unresponsive to these treatments [4].

Indeed, the clinical diagnosis and treatment of depression based on the Diagnostic and Statistical Manual of Mental Disorders (DSM) or the wide-ranging International Statistical Classification of Diseases and Related Health Problems (ICD) have focused on observable behaviors (signs) and self-reported feelings and thoughts (symptoms). Classifying mental disorders according to clinical signs and symptoms has led to a limitation in reflecting the underlying pathophysiology, and to heterogeneity within groups diagnosed with the same psychiatric disease [5]. Thus, attempts have emerged to suggest the novel classification of mental disorders that reflects biological mechanisms, such as Research Domain Criteria (RDoC) and biological classification of mental disorders (BeCOME) study [6,7]. Furthermore, many studies have aimed to identify the pathomechanism of depression to overcome the limitations of other existing tools for its diagnosis and treatment.

In addition to the well-known monoaminergic neurotransmitter dysfunction, altered hypothalamic-pituitary-adrenal (HPA) axis activity, dysfunctional brain network activity, impaired neurotrophic factor signaling, and neuroinflammation have been implicated in depression and studied for potential diagnostic biomarkers and therapeutic targets [8,9,10]. Additionally, changes in brain structure [11,12], gastrointestinal factors [13,14], oxidative stress [15], and endocannabinoid system components [16] have also been implicated in depression [17]. In addition, correlation studies for the aforementioned biomarkers such as inflammatory factors and brain structural changes also have been conducted in depression [18,19]. Family, twin, and adoption studies suggest that genetic factors account for 30–40% of the variance in depression risk [20], but early genome-wide association studies (GWASs) failed to identify genetic variants strongly associated with depression, suggesting that genetic susceptibility is mediated by heterogeneous combinations of risk alleles [21,22,23]. However, recent GWASs have identified several genetic loci reproducibly associated with depression [24,25,26,27,28].

The remaining 60–70% of the variation in depression risk appears to be determined by environmental factors [29]. Environmental stressors such as physical, emotional, and sexual abuse, social rejection, and other early adverse experiences and stressful life events such as the death of a loved one, illness, injury, disability, and functional decline are demonstrated risk factors for depression [30,31,32]. Individual variations in susceptibility to such stimuli may be explained in part by genetic factors. Indeed, a gene-environment interaction model positing that penetrant and complex genetic predispositions interact with environmental factors to determine depression susceptibility is now widely accepted [33].

In this gene-environmental interaction model, epigenetic mechanisms act as a bridge between genes and environmental factors [34]. Epigenetics refers to “heritable, but reversible, regulation of various genomic functions mediated principally through changes in DNA methylation and chromatin structure” [35]. Thus, epigenetic mechanisms are the processes by which various types of cells within the same organism acquire unique transcriptional properties and functions during development [36]. This dynamic and reversible process also contributes to the transcriptional plasticity manifested by the neurons and glia in the brain. Therefore, it is associated with learning and memory, age-related neurodegeneration, cognitive and behavioral effects of early experiences, repeated drug exposure, chronic stress, prolonged changes in nutritional status, and exposure to environmental toxins [37]. The functional analyses of DNA methylation quantitative trait locus (meQTL) and non-coding RNA (ncRNA) in depression-associated single nucleotide polymorphisms (SNPs) revealed that alterations in DNA methylation and ncRNAs interact with genetic factors in depression, which underscores the importance of epigenetic regulation for depression [38]. Thus, the present review provides an overview of the impact of histone deacetylation on the pathophysiology of depression and the therapeutic potential of its modulation.

## 2. Histone Acetylation

Dynamic acetylation and deacetylation of histone lysine (Lys) residues control the packaging of genomic DNA, thereby influencing DNA replication, transcription, DNA repair, and cell cycle progression [39]. Histone acetyltransferase enzymes (HATs) catalyze the transfer of acetyl groups from acetyl CoA to the ε-amino groups of Lys residues within histones [40], while histone deacetylases (HDACs) remove these acetyl groups [41]. Thus, the balance between HAT and HDAC activities determines the net histone acetylation status of the genome. By dynamically modulating the interaction between histones and DNA at the local level, histone acetylation regulates the accessibility of gene promoters to various binding factors such as transcription factors. In addition, acetylation/deacetylation of non-histone proteins modulated by HATs and HDACs also regulates diverse cellular functions [42].

## 3. Histone Deacetylase (HDAC) Families and Classes

Human HDACs are traditionally divided into two families, the Zn^2+^-dependent amidohydrolases including class I, II, and IV HDACs and the NAD^+^-dependent class III SIRT enzymes (Table 1). To date, 18 HDACs have been identified in humans and are grouped by sequence homology and domain organization [43]. Class I HDACs share structural homology with the yeast transcriptional regulator Rpd3 and typically act as the catalytic subunit within a complex of cognate corepressors to inhibit transcription in the cell nucleus [44]. HDAC1 and 2 are present in NuRD, Sin3, NODE, CoREST, and MiDAC complexes, while HDAC3 is a component of SMRT and NCoR corepressor complexes [45,46]. In contrast, HDAC8 can function independently without forming a multiprotein complex [47].

Class II HDACs are highly homologous to yeast Hda1 and are subdivided into two groups [48]. Class IIa HDACs 4, 5, 7, and 9 each have a single catalytic domain and a unique adaptor domain including a transcription factor MEF2-binding motif [49], while class IIb HDACs 6 and 10 contain two catalytic domains, a ubiquitin-binding zinc finger domain and a leucine-rich repeat domain [50,51,52,53,54]. In contrast to class I HDACs, which are exclusively localized in the nucleus, class II enzymes can shuttle between the cytoplasm and nucleus in response to various regulatory cues [49].

HDAC11, a homolog of yeast Hos3, is the only member of Class IV [55]. It is primarily expressed in the brain, skeletal muscle, heart, testis, and kidney, suggesting specific functions in development, inflammation, metabolism [55].

Class III HDACs are homologous to yeast Sir2. Like other HDACs, Class III members are involved in transcriptional silencing but have a deoxyhypusine synthase-like NAD/FAD-binding domain clearly distinct from the catalytic domains of other HDAC classes [56]. Seven Sir2-like proteins (SIRT1-7), referred to as sirtuins, have been identified in humans [57]. These sirtuins possess additional domain(s) such as a mono-ADP-ribosyltransferase domain. SIRT1 has the strongest histone deacetylase activity among sirtuins, while SIRT5 shows weak deacetylase activity but robust lysine desuccinylase and demalonylase activities [58]. These enzymes are differentially localized to the nucleus (SIRT1, 2, 3, 6, and 7), cytoplasm (SIRT1 and 2), and mitochondria (SIRT3, 4, and 5) [43].

## 4. HDAC and Depression

Among the epigenetic mechanisms, the most well-studied for contributions to depression are DNA methylation mediated by DNA methyltransferases (DNMTs) and histone post-transcriptional modifications (PTMs), including acetylation/deacetylation. Associations between depression and DNA methylation have been suggested in many studies. For example, increased *DNMT3A* levels were found in the nucleus accumbens (NAc), the limbic region regulating reward behavior, in the postmortem brains of depressed patients, and in animal models of depression [59,60]. Data on DNA methylation age (DNAm age) derived from blood and brain tissues indicate that patients with depression displayed higher levels of epigenetic aging than those with normal subjects [61].

Along with DNA methylation, histone acetylation via HAT and deacetylation via HDAC are reported to be crucial for long-term stress adaptation and responses to antidepressant therapy [34]. Further, several studies have suggested a relationship between depression and histone deacetylation. Chronic social defeat stress transiently suppressed histone acetylation in the NAc of mice [62], while HDAC inhibition exerted antidepressant-like effects in animal models of stress-induced depression [62,63,64,65,66]. Moreover, the expression levels of *HDAC2* and *HDAC5* mRNAs in peripheral white blood cells were elevated in depressed patients compared to healthy controls [67]. Singh et al. [68,69] also reported the association between depression and HDAC6, which contributes to the stabilization of microtubules in the brain by regulating acetylation of α-tubulin. Interestingly, the effects of early-life stress (e.g., maternal separation) and subsequent environmental enrichment on depressive behavior and HDAC/DNMT activities in the hippocampus and prefrontal cortex (PFC) are sex-dependent, which supports sex differences in the prevalence of depression [70].

Diverse reports have suggested that sirtuins, categorized as class III HDACs, play several roles in the mammalian brain, such as modulating brain structure through axon elongation, outgrowth of neurites, and dendritic branching [71]. Among such sirtuin proteins, SIRT1 is associated with high-order brain function including synaptic plasticity and memory formation [72]. As a result of studies based on these reported functions of SIRT1, many researchers have demonstrated the relationship between SIRT1 and depression. For example, the expression of *SIRT1* in peripheral blood was downregulated in depressed patients compared to healthy controls [73]. Furthermore, these results were reproduced in animal studies; altered activity of SIRT1 in the hippocampus and the NAc provoked depressive-like behaviors in animal models of depression [74,75].

## 5. HDAC and the Hypothalamic-Pituitary-Adrenal (HPA) Axis

From the epigenetic perspective, stress is considered to be an important factor in the etiology of stress-related disorders such as depression and anxiety [76]. When exposed to social and physical stressors, the paraventricular nucleus (PVN) of the hypothalamus is stimulated to secrete both corticotrophin-releasing hormone (CRH) and arginine vasopressin (AVP) which stimulate the release of adrenocorticotropic hormone (ACTH) in the pituitary gland. Consequently, mainly cortisol in humans and corticosterone in rodents are produced in the adrenal cortex and released into the bloodstream, exerting their effects through glucocorticoid receptors (GRs) in each tissue. The activation of GRs in the PVN of the hypothalamus and pituitary corticotroph cells inhibits the hypothalamic release of CRH and AVP and contributes to the negative feedback regulation of the HPA axis [77,78,79,80]. Additionally, the hippocampus can contribute to feedback regulation of the HPA axis through GR signaling [81]. This regulation is important in handling challenging situations and maintaining homeostasis (Figure 1).

Stress, especially in chronic or developmentally critical periods (i.e., prenatal and postnatal periods), influences various epigenetic mechanisms including DNA methylations and histone modifications, leading to structural and regulatory changes and fine-tunes the neural circuitry [82,83,84]. For example, researchers reported that early-life stress influences HDAC expression in the mouse brain [85,86]. Given that the HPA axis is one of the main stress responses, many researchers investigated the epigenetic regulation of the HPA axis in depression and identified the indirect effects of HDACs on the HPA axis. Murgatroyd et al. [87] focused on AVP which was reported to be important in the regulation of mood behaviors [88]. The authors demonstrated that early-life stress, represented by maternal deprivation, modulated AVP expression dynamically in the PVN of the hypothalamus initially through methyl CpG binding protein 2 (MeCP2) phosphorylation and later by AVP enhancer hypomethylation [87]. Considering that MeCP2 forms a complex consisting of HDAC and DNMT, consequently inducing gene silencing, HDAC is considered as a modulator of the HPA axis (Figure 2).

Unlike AVP, CRH expression in the hypothalamus, another component of the HPA axis, was not changed by maternal deprivation [87]. However, GR expression in the hippocampus was influenced under early-life stress through epigenetic mechanisms. Maternal deprivation affected DNA methylation status in the promoter of *GR* exons in the hippocampus, which mediates the recruitment of HDAC-containing repressor complexes (e.g., HDAC5) to hypermethylated loci [89,90]. These effects of early-life stress on hippocampal GR were reversed by HDAC inhibitors such as trichostatin A.

## 6. HDAC and Brain-Derived Neurotrophic Factor

Brain-derived neurotrophic factor (BDNF) is a critical ligand guiding neurodevelopment and the ongoing neuroplastic processes required for behavioral adaptation, such as neurogenesis, synaptic plasticity, dendritic arborization, and pruning, and dendritic spine maturation [91,92]. Antidepressants and exercise increase endogenous BDNF in rodents, resulting in enhanced neurogenesis, reduced neuronal apoptosis, and inhibition of stress-induced depressive-like behaviors [92], while reduced BDNF is associated with depression as well as other neuropsychiatric and neurologic diseases such as Parkinson’s disease and Alzheimer’s disease [91]. Further, lower BDNF levels are observed in the PFC and the hippocampus of suicide victims compared to non-victims of suicide with or without depression [93].

Expression of BDNF is influenced by environmental stimuli via histone modification at different promoter sites in distinct brain regions especially during development [94]. Prenatal stress exposure was reported to increase HDAC expression and decrease BDNF expression in the hippocampus, resulting in anxiety- and depression-like behaviors [95]. In addition to prenatal stress, early postnatal stress also induced changes in histone modification and an increase of HDAC in the hippocampus, leading to changes in BDNF expression and behavior in rodents [92]. Not only during development but stress during adulthood also up-regulated MeCP2 levels at the *Bdnf* promoter and *Hdac5* expression in the hippocampus [96]. On the other hand, antidepressants and HDAC inhibitors (e.g., sodium butyrate, trichostatin A, and valproic acid) increased BDNF expression and it was associated with reduced DNA methylation and histone deacetylation around the *Bdnf* promoter region [97,98,99]. Similar to other HDACs, SIRT1 also can regulate BDNF expression through interaction with MeCP2 [100].

## 7. HDAC and Neuronal Plasticity

Experience-dependent neuronal plasticity, characterized by sustained changes in synaptic structure and strength, is the neurocellular basis for sensing, adapting, and responding to environmental changes, including stress [101,102]. Thus, it is not surprising that aberrant synaptic plasticity is associated with the pathophysiology of depression. Indeed, both preclinical models of depression and depressed patients exhibit abnormalities in factors that regulate synaptic plasticity [33,103,104]. One of the strongest factors disrupting normal neuronal plasticity is chronic stress, and severe or chronic stress can reduce the capacity of the brain to respond and adapt to stress, resulting in depression [102,105]. Stressors activate the HPA axis and consequently increase circulating glucocorticoid levels. Chronically elevated glucocorticoid decreases synaptic number, impairs plasticity, and leads to neuronal atrophy, resulting in disrupted neural circuitry within and among regions regulating mood, executive function, and cognition [101]. Moreover, glucocorticoid can alter gene transcription via epigenetic regulation of the GR [89].

Abnormal histone acetylation due to the imbalance between HAT and HDAC activities can also impair synaptic plasticity, thereby reducing cognitive capacity and inducing abnormal behaviors. For instance, histone lysine acetylation can enhance neuronal plasticity while activation of HDAC and concomitant deacetylation can impair neuronal plasticity [106]. The administration of the non-selective HDAC inhibitor sodium butyrate enhanced histone acetylation and long-term potentiation (LTP), a form of synaptic plasticity strongly implicated in learning and memory, and improved memory performance [107]. Conversely, HDAC2 overexpression reduced synaptic number and synaptic plasticity, resulting in long-lasting neural circuit abnormalities and memory impairment. These changes may occur via the inactivation of activity-dependent genes involved in synaptic plasticity. Further, these effects were reversed by the HDAC inhibitor suberoylanilide hydroxamic acid (SAHA) [108].

In addition to HDAC2, HDAC4 is also implicated in the regulation of neuronal plasticity. HDAC4 is a transcriptional repressor that can translocate from the neuronal cytoplasm to the nucleus, bind chromatin, and suppress the expression of transcription factors critical for synaptic plasticity and information processing such as myocyte enhancer factor 2A (MEF2A) and cAMP response element-binding protein (CREB) [109,110,111]. Brain-specific HDAC4 knockout in mice impaired hippocampus-dependent memory and long-term synaptic plasticity [112]. Chronic cocaine-induced promoter-specific change in HDAC3, which is known as a negative regulator of memory formation, in the NAc and interfering HDAC3 activity restored cocaine-induced synaptic plasticity [113]. In addition, SIRT1 knockout mice also exhibited impaired memory and hippocampal plasticity [72]. Taken together, these findings indicate that appropriate HDAC function is essential for synaptic and neuronal plasticity and that an abnormal shift in histone acetylation status can result in impaired neural plasticity and behavioral dysfunction.

## 8. Molecular Diagnosis of Depression: An Epigenetic Perspective

Studies on the pathophysiology of depression have identified several promising prognostic and diagnostic biomarkers, including factors associated with the HPA axis (e.g., CRH, ACTH, and cortisol), inflammatory factors (e.g., tumor necrosis factor (TNF)-α, interleukin (IL)-1β, IL-6, and C-reactive protein (CRP)), neurotrophic factors (e.g., BDNF and glial cell line-derived neurotrophic factor (GDNF)), insulin-like growth factor 1 (IGF-1), and changes in the area or volume of the hippocampus, amygdala, and PFC [17,114,115]. According to Kennis et al. [17], only cortisol in saliva was a significant biomarker for the onset/relapse/recurrence of depression, but careful interpretation is needed given the methodological heterogeneity among included studies.

In addition, several studies have identified the genes encoding the serotonin transporter (SLC6A4) [116,117], IL-1β [118,119,120], and FK506 binding protein 5 (FKBP5 or FKBP-51) [116,121] as potential genetic biomarkers for depression. The genetic loci related to depression (e.g., SNPs in *LHPP*, *SIRT1* region) have also been revealed although there are differences between studies [24,26]. Furthermore, there are attempts to identify blood gene expression biomarkers and provide predictive information as well as precise and personalized diagnosis and treatment for depression [117,122]. Recently, researchers have attempted to integrate functional neuroimaging and genetic data (neuroimaging genetics) for depression. Buch et al. [123] found that polymorphisms of the serotonin transporter (5-HTTLPR) and *BDNF* genes were associated with structural and functional changes in the anterior cingulate cortex, amygdala, and hippocampus, regions of the mesocorticolimbic reward circuit strongly associated with behaviors impaired in depression [124]. These results provide a novel diagnostic strategy for depression and imply that genetic factors contribute to depression by modulating brain structure and function.

The diagnostic biomarkers associated with epigenetic regulation also have been attracted attention in various diseases including neuropsychiatric diseases [121,125]. For instance, an epigenome-wide association study by Jovanova et al. [126] identified the methylation of 3 CpG islands in blood associated with depression. Moreover, hypermethylation of *BDNF* and *SLC6A4* genes have been found in depressed patients [127]. The local regions of histone acetylation may also serve as possible biomarkers for depression, as both animal and human postmortem studies have reported associations between histone modifications in brain tissue and depression. In addition, histone H3 lysine 27 trimethylation (H3K27me3) at the *BDNF* gene promoter IV of peripheral blood was downregulated in an antidepressant-responder group compared to a non-responder group [128]. Also, HDAC5 activity was significantly higher in peripheral leukocytes from drug-free depressive patients and normalized by antidepressant treatment [129]. The plasma levels of acetyl-L-carnitine (LAC), an acetylating agent that can pass through the blood–brain barrier, were decreased in depressed patients compared to control, where the degree of reduction in LAC was much greater in patients with treatment-resistant depression [130].

In recent years, diverse attempts have been conducted to visualize epigenetic factors and utilize them for diagnosis. For example, a positron emission tomography (PET) imaging study in human using [^11^C] Martinostat, the only selective tracer for class I/IIb HDAC in the central nervous system [131,132], demonstrated that [^11^C] Martinostat uptake in the dorsolateral PFC of patients with schizophrenia/schizoaffective disorder was lower compared to those of healthy controls, which is inconsistent with the results of postmortem studies [133]. Additionally, low [^11^C] Martinostat uptake was observed in the frontolimbic areas of patients with bipolar disorder compared with healthy controls [134]. Since no visualization studies have been published related to depression yet and it is still in its infancy, many additional studies are expected to be needed to apply them to a depression diagnosis.

## 9. Molecular Therapeutics of Depression: An Epigenetic Perspective

The current first-line therapies for depression are tricyclic antidepressants (TCAs), MAO inhibitors, and SSRIs, all of which target the dysfunction of monoaminergic transmission [115]. However, classical antidepressants such as TCAs (e.g., imipramine) and SSRIs (e.g., paroxetine, fluoxetine, and escitalopram) not only bind to monoamine transporters but also have indirect effects on both DNA methylation and histone PTM [135]. For example, the reduced DNA methylation at the *Cr**h* promoter and increased *Crh* mRNA expression in chronic social defeat stress-induced depression were reversed by chronic imipramine administration [136]. Additionally, the SSRI paroxetine was reported to inhibit DNMTs [98]. Chronic antidepressant administration was also found to increase acetylated histone H3 (AcH3) levels by reducing HDAC expression in several brain regions, including the NAc [137].

DNMT inhibitors are not approved as antidepressant drugs despite their documented antidepressant effects because modulation of global brain methylation can cause cognitive deficits [135]. However, HDAC inhibitors have been examined as novel therapeutics for treatment-resistant depression [34,138,139], and numerous preclinical studies have reported that various HDAC inhibitors exert antidepressant-like effects in animal models of stress-induced depression [62,63,66,140,141,142] (Table 2). In addition to the antidepressant effect, HDAC inhibitors promoted neuronal rewiring and recovery of motor functions after traumatic brain injury [143]. Also, HDAC inhibitors such as sodium butyrate and SAHA enhanced cognitive function, which may provide therapeutic options for depression that accompanies cognitive impairment [144,145,146]. A recent drug repositioning study for precise/personalized medicine in depression using bioinformatic analyses revealed that HDAC inhibitors such as trichostatin A and valproic acid as a new potential antidepressant drug [117].

While these results support the potential of HDAC inhibitors as novel therapeutic drugs for depression, their use in clinical practice is limited by severe side effects including thrombocytopenia and neutropenia [147,148]. Although several HDAC inhibitors, including vorinostat (SAHA), belinostat, panobinostat (LBH-589), romidepsin (FK2280), have been approved by the Food and Drug Agency (United Stated), the clinical application of these drugs is limited to certain forms of cancers (e.g., T-cell lymphoma and multiple myeloma) [149] and to date, there is no clinical trial evaluating the antidepressant effect of HDAC inhibitors in depression.

Apart from HDAC inhibitors, the acetylating agent LAC also has been reported to be a potential antidepressant that is mediated by neurotransmitter regulations such as serotonin and epigenetic regulation of key genes important for synaptic plasticity (e.g., *BDNF* and metabotropic glutamate receptor of class-2 (*mGlu2*)) [130]. Lactate, a metabolite produced by exercise, induced resilience to social defeat stress and reversed social avoidance behavior and anxiety by modulating the activity of HDAC2 and HDAC3 [150]. In addition, dihydrocaffeic acid (DHCA) and malvidin-3′-O-glucoside (Mal-gluc) induced a resilient state against social stress and attenuated depressive behaviors via epigenetic regulation [151]. In particular, Mal-gluc mediates the increase in histone acetylation of the *Rac1* gene regulatory sequence through HDAC2 inhibition, and as a result, the modulation of synaptic plasticity occurs.

## 10. Conclusions

Depression is a common and disabling psychiatric disease with high recurrence rates and heterogeneous clinical manifestations, adding to treatment complexity and suggesting that depression is not a unitary disease entity. Indeed, numerous pathomechanisms likely contribute to depression, including abnormal epigenetic changes. Environmental stressors are the primary risk factors for depression, supporting contributions of epigenetic mechanisms to disease pathogenesis and progression. In this review, we summarized the latest knowledge on potential epigenetic mechanisms, especially histone acetylation/deacetylation, underlying disease pathophysiology, the utility of epigenetic markers for diagnosis, and the potential of epigenetic modulators, especially HDAC inhibitors, as therapeutics. Recent studies have shown that HDAC inhibition can upregulate BDNF expression, resulting in enhanced neural/synaptic plasticity, and exert an antidepressant-like effect on behavior. Conventional antidepressants targeting monoaminergic neurotransmission also modulate epigenetic mechanisms, further supporting the contributions of epigenetic dysregulation to the pathophysiology of depression. Thus, HDACs can be regarded as novel diagnostic and therapeutic targets for depression. However, further studies are needed to develop safe and effective HDAC inhibitors for clinical use.

## Figures and Tables

**Figure 1 ijms-22-05398-f001:**
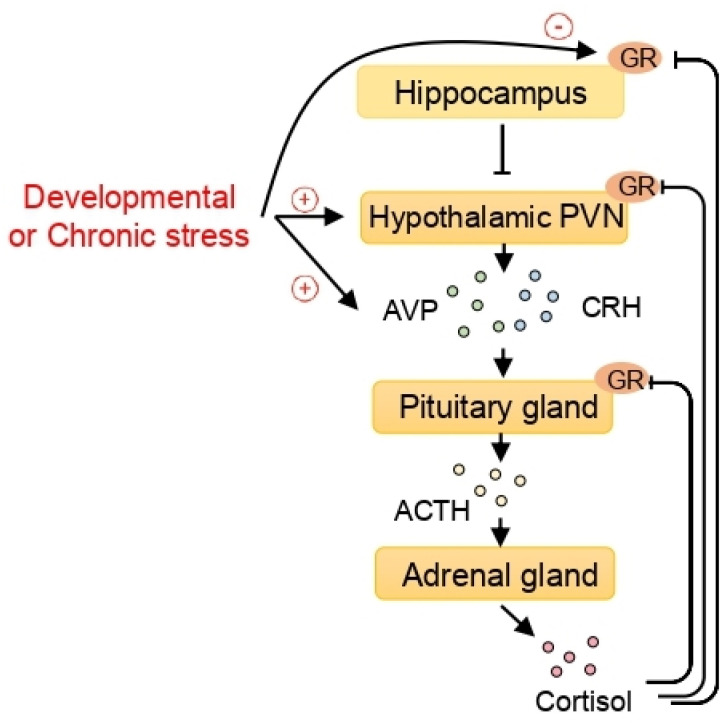
The epigenetic effect of stress on the hypothalamic-pituitary-adrenal (HPA) axis and the epigenetic regulation of arginine vasopressin (AVP) expression. When exposed to stress, corticotrophin-releasing hormone (CRH) and AVP, released from the paraventricular nucleus (PVN) of the hypothalamus, stimulate the pituitary gland to secrete adrenocorticotropic hormone (ACTH). The adrenal glands, activated by ACTH, secrete cortisol. Cortisol exerts its function by binding to the glucocorticoid receptors (GRs). In turn, the GRs in the pituitary gland, the hypothalamic PVN, and the hippocampus play important roles in the feedback regulation of the HPA axis. Developmental or chronic stress, which can program the HPA axis, increases AVP expression and decreases hippocampal GR through epigenetic mechanisms including histone deacetylases (HDACs).

**Figure 2 ijms-22-05398-f002:**
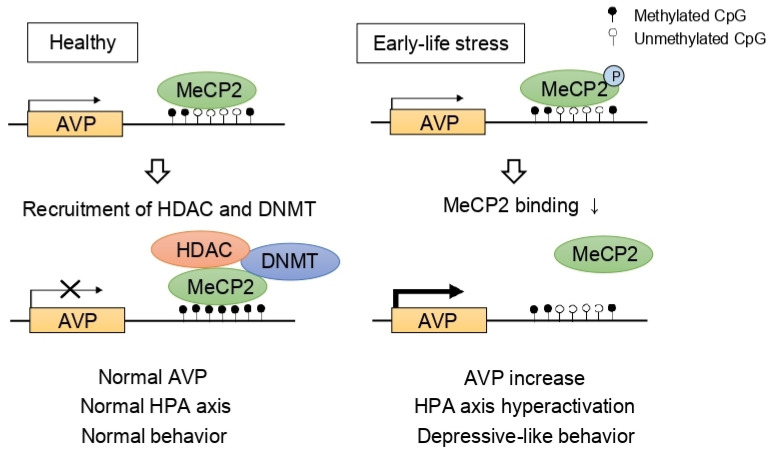
Epigenetic programming of arginine vasopressin (AVP)**.** In normal conditions, AVP expression is repressed by methyl CpG binding protein 2 (MeCP2), DNA methyltransferase (DNMT), and histone deacetylase (HDAC) complex. However, early-life stress induces MeCP2 phosphorylation, inhibiting the recruitment of DNMT and HDAC consequently leading to hypomethylation at the AVP enhancer. As a result, increased AVP levels contribute to hyperactivation of the HPA axis and depressive-like behaviors.

**Table 1 ijms-22-05398-t001:** HDAC classification.

Class	Protein (*S. cerevisiae*)	Protein (Human)	Subcellular Localization
Class I	Rpd3	HDAC1	Nucleus
		HDAC2	Nucleus
		HDAC3	Nucleus
		HDAC8	Nucleus
Class IIa	Hda1	HDAC4	Nucleus/cytoplasm
		HDAC5	Nucleus/cytoplasm
		HDAC7	Nucleus/cytoplasm
		HDAC9	Nucleus/cytoplasm
Class IIb	Hda1	HDAC6	Cytoplasm
		HDAC10	Cytoplasm
Class IV	Hos3	HDAC11	Nucleus/cytoplasm
Class III	Sir2	SIRT1	Nucleus/cytoplasm
		SIRT2	Nucleus/cytoplasm
		SIRT3	Nucleus/mitochondria
		SIRT4	Mitochondria
		SIRT5	Mitochondria
		SIRT6	Nucleus
		SIRT7	Nucleus

**Table 2 ijms-22-05398-t002:** Summary of the antidepressant actions of HDAC inhibitor in animal model.

HDAC Inhibitor	Animal Model	Measurement of Antidepressant Effect	Molecular Mechanisms of Action	Ref.
MS-275	Chronic social defeat stress	Social avoidance, sucrose preference, FST	acH3 ↑ in the NAc	[62]
	Chronic social defeat stress	Sucrose preference test, social avoidance (combined with social enrichment)	acH3 ↑ in the hippocampus	[63]
	Chronic social defeat stress	Social avoidance, FST	acH3 ↑ in the mPFC	[64]
	Chronic social defeat stress	Social avoidance	Rac1 ↑ in the NAc synapse structural plasticity normalization	[141]
SAHA	Chronic social defeat stress	Social avoidance, sucrose preference, FST	acH3 ↑ in the NAc	[62]
	Chronic unpredictable mild stress	Social interaction, sucrose preference test, novelty-suppressed test, FST	HDAC2 inhibition, *Gdnf* ↑ in the NAc	[140]
Sodium butyrate	Behavioral despair paradigm	TST	acH3 ↑ in the hippocampus, *Bdnf*↑ in the frontal cortex	[65]
	Chronic social defeat stress	Social avoidance	HDAC5 inhibition, acH3 ↑ in *Bdnf* gene P3, P4 promotor	[66]
	Chronic restraint stress	Sucrose preference test, Light/dark test, TST, FST	HDAC2 ↑, pCREB ↑, AcH3 ↑, BDNF ↑ in the hippocampus	[142]

BDNF, brain-derived neurotrophic factor; CREB, cAMP response element-binding protein; FST, forced swim test; GDNF, glial cell-derived neurotrophic factor; HDAC, histone deacetylase; mPFC, medial prefrontal cortex; NAc, nucleus accumbens; Rac1, Rac family small GTPase 1; SAHA, suberoylanilide hydroxamic acid; TST, tail suspension test; ↑ increase.

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
