# Peer review of "Epigenetic Targeting of Histone Deacetylases in Diagnostics and Treatment of Depression"

_ijms, 2021, doi:10.3390/ijms22105398_

Round 1

Reviewer 1 Report

  1. Authors conlude in the end of abstract: "This review descibes recent advances ...." Authors should stress in the manuscript novel aspects and studies which have been done after 2018 year, when the last one similar review was published. (P.Misztak et al. 2018; Fuchikami et atl. 2016).
  2. There is no description of the latest reserach. Literature should be updated.
  3. Please create the table which include: HDIs/model/molecular mechanism of action in depression with citations. All studies should be summarizied in the manuscript in the more visible way.

Author Response

We thank the reviewer for the positive review and useful comments.

1. What we focused on differentiation from previous papers in this manuscript was the description of the epigenetic regulation (especially HDAC) of the previously known pathophysiology of depression (HPA axis, brain-derived neurotrophic factor and neuronal plasticity). In addition, beyond the existing papers focused mainly on the potential of HDAC inhibitors as the therapeutic agent, we tried to introduced several studies to identify epigenetic diagnostic biomarkers that can be used in real depressed patients. However, we agree with the reviewer's opinion that it is necessary to emphasize the differences from similar reviews and update the latest research. Therefore, we added several latest research reports on the diagnosis and treatment of depression from an epigenetic perspective in the main text and revised the manuscript to address effectively the importance of HDACs in depression.

In the introduction part, a paragraph was newly added to introduce the biological approachs to depression and several studies to overcome the limitations of the existing diagnosis and treatment of depression (RDoC and beCOME study; 2nd paragraph). In the ‘molecular diagnosis of depression: an epigenetic perspective’ part, 1st sentence in 3rd paragraph was revised to clarify the topic (line 355-356). In addition, we introduced some studies that aimed to not only diagnose but also predict the prognosis and recommend effective drugs for depression, beyond the previous simple biomarker screening reports in the ‘molecular diagnosis of depression: an epigenetic perspective’ part, and ‘molecular therapeutics of depression: an epigenetic perspective’ part (line 345-347, 419-422). Other novel findings in diagnosis and treatments of depression with epigenetic perspective were added in the revised manuscript to inform recent advances of diagnosis and treatment of depression (line 366-369, 417-419, 425-441; detail in comment 2)

2. This is a good point. Therefore, the more recent studies were updated in the manuscript.

In the 3rd paragraph of introduction section, ‘endocannabinoid system components’ were added to the list of factors implicated in depression (line 60). Also, several studies were newly introduced in the revised manuscript to describe appropriately the relationship between the interactions of previously revealed depression-related factors and depression (line 61-63). In addition, in the 5th paragraph of introduction section, we added the latest multi-layered (genetic, epigenetic, transcriptomic) approach to depression that supports epigenetic regulation of depression (line 87-90).

In the ‘HDAC and depression’ section, we newly added some cases in the edited manuscript to introduce diverse examples; recent studies which describe epigenetic changes (DNA methylation age) in depressed patients (line 151-153) and studies for HDAC6, alpha-tubulin and depression to the 2nd paragraph (line 162-169). We also introduced a study that mentioned sex differences in the epigenetic regulation of depression (line 169-173). In addition, we added the recent study which showed synaptic plasticity regulation of HDAC3 to the 3rd paragraph of ‘HDAC and neural plasticity’ section (line 324-326).

The new possible biomarker, acetyl-carnitine (LAC) were added to the 3rd paragraph of ‘Molecular diagnosis of depression: an epigenetic perspective’ section (line 366-369). In the ‘Molecular therapeutics of depression: an epigenetic perspective’ section, the effects of enhancing cognitive function of HDAC inhibitor which can act synergistically with antidepressant effect were newly described (line 417-419) and new possible epigenetic agents for depression such as LAC, Lactate, dihydrocaffeic acid (DHCA), and malvidin-3’-O-glucoside (Mal-gluc) were also introduced in the revised manuscript (line 431-441).

3. This is a good suggestion. In response to your opinion, we added new table (Table 2) in the revised manuscript to improve readability.

Reviewer 2 Report

Park, Kim, Ahn and Ryu present here an interesting review of the role of histone deacetylase in depression, covering both biological basis and clinical utility. The topic is certainly very interesting, and the writing is high quality, however I currently do not feel that the content is covered in as much details as it could be. Notably a substantial number of the citations, particularly in the opening third of the manuscript, are to other review articles. Additionally, in spite of the authors stated aim of describing “recent advances” in the field, surprising few of the papers cited are original research articles from the last 3-5 years. While the concept of the review is therefore strong, I feel that more detailed discussion of recent findings is required for it to fulfil its potential.

Additional minor comments:

The “HDAC families and classes” section, by necessity, includes a lot of complex detail. Would it be more reader friendly if at least some of this was instead presented in a table or figure?

On line 198, the description of control individuals as “non-suicides” is inappropriate. “non-victims of suicide” or “those who were not victims of suicide”?

Finally, while the quality of the English is general high, there are some mistakes and so an additional proof read would be good (e.g. line 140 “HDAC and the HPA axis”, line 219 “…stress and severe or…”, line 248 punctuation is required between “depression” and “an”, line 263 “Recently, researchers…”).

Author Response

Response to major comments

We thank the referee for the positive overall evaluation and the constructive comments to extend some points in the paper. The reviewer’s comments are closely related to #2 question from #1 referee. As noted above, we have now added diverse latest research on the diagnosis and treatment of depression from an epigenetic perspective in the revised manuscript to introduce effectively several cases of “recent advances” in the field.

In the introduction part, a paragraph was newly added to introduce the background of the biological approach to depression and several studies to overcome the limitations of the existing diagnosis and treatment of depression (RDoC and beCOME study; 2nd paragraph).

In the 3rd paragraph of introduction section, ‘endocannabinoid system components’ were added to the list of factors implicated in depression (line 60). Also, several studies were added in the revised manuscript to describe the relationship between the interactions of previously revealed depression-related factors and depression (line 61-63). In addition, in the 5th paragraph of introduction section, we introduced the latest multi-layered (genetic, epigenetic, transcriptomic) approach to depression that supports epigenetic regulation of depression (line 87-90).

In the ‘HDAC and depression’ section, we newly added some cases in the edited manuscript to introduce diverse examples; recent studies which describe epigenetic changes (DNA methylation age) in depressed patients (line 151-153) and studies for HDAC6, alpha-tubulin and depression to the 2nd paragraph (line 162-169). We also introduced a study that mentioned sex differences in the epigenetic regulation of depression (line 169-173). In addition, we added the recent study which showed synaptic plasticity regulation of HDAC3 to the 3rd paragraph of ‘HDAC and neural plasticity’ section (line 324-326).

In addition, we introduced diverse studies that aimed to not only diagnose but also predict the prognosis and recommend effective drugs for depression, beyond the previous simple biomarker screening studies in the ‘molecular diagnosis of depression: an epigenetic perspective’ part, and ‘molecular therapeutics of depression: an epigenetic perspective’ part (line 345-347, 419-422). Other novel findings in diagnosis and treatments of depression with epigenetic perspective were added to the revised manuscript to introduce recent advances on diagnosis and treatment of depression (line 366-369, 417-419, 425-441). We also added statements in the main text that contain limited uses of several HDAC inhibitors currently approved by the FDA as well as the current status of clinical trials involving HDAC inhibitors in depression (line 425-430).

Response to minor comments

1) This is a good suggestion. We also think a table is worthwhile to show clearly classification of HDAC proteins. Therefore, we added newly Table 1 in the revised manuscript.

2) Thank you for catching this! We have corrected it in the edited manuscript.

3) We also thank you for pointing out these errors, which have been corrected.

Round 2

Reviewer 1 Report

I'm satisfied with the authors' response and corrections.

Reviewer 2 Report

The authors have done an admiral job of revising their manuscript in a short amount of time. I feel that my concerns have been addressed, and that this is now suitable for publication.